# Simultaneous Detection and Grading of Prostate Cancer in Multi-Parametric MRI

**Coen de Vente**[1,2]                                    C.W.D.VENTE@STUDENT.TUE.NL

**Pieter Vos**[2]                                          PIETER.VOS@PHILIPS.COM

**Josien Pluim**[1]                                        J.PLUIM@TUE.NL

**Mitko Veta**[1]                                          M.VETA@TUE.NL

[1] *Department of Biomedical Engineering, Eindhoven University of Technology, The Netherlands*

[2] *Philips Research, Eindhoven, The Netherlands*

## 1. Introduction

If there are indications that a patient has prostate cancer (PCa), a biopsy is ordered (Frankel et al., 2003). Gleason Grade Group ($GGG$), a predictor of pathological stage and oncological outcomes, (scale 1-5) is determined from the microscopic analysis of the biopsy samples (Epstein et al., 2016). With $GGG$ 1 no treatment is usually needed and $GGG$ 5 is the most severe type of PCa. Lesion grading from multi-parametric MRI (mpMRI) can decrease the number of biopsies. This is currently being done with the PIRADSv2 system. However, its inter-rater reliability remains moderate (Baldisserotto et al., 2016). The goal of the current study is to develop a more objective grading method from mpMRI.

In previous works, methods have been developed to detect PCa from mpMRI (Kohl et al., 2017; Tsehay et al., 2017). Other methods classify given lesions only as clinically significant or insignificant (Armato et al., 2018; Liu et al., 2017; Yuan et al., 2018; Seah et al., 2017). Furthermore, techniques have been developed to grade suspicious regions indicated by radiologists (Armato et al., 2018; Jensen et al., 2019; Abraham and Nair, 2019). In this work, we propose an approach to perform both detection and grading. Furthermore, multiple methods to grade $GGG$s in mpMRI are evaluated, of which the best performing method, to best of our knowledge, has not previously been applied to $GGG$ scoring.

## 2. Materials and methods

**Dataset:** We used the ProstateX-2 Challenge (PX-2) dataset for training, validation and testing. The challenge training set contains 99 patients and 112 lesions. The challenge test set contains 63 patients and 70 lesions. For every patient, mpMRI images were given. From the four types of scan data, we used T2-weighted (T2-w) and apparent diffusion coefficient (ADC). The challenge provided coordinates and $GGG$s of the training set lesions. Pathological analysis of MR targeted biopsy tissue provided the ground truth for the $GGG$ scores. The aim of PX-2 was to assign a $GGG$ to the lesions, given their coordinates. We manually delineated the lesions with in-house software that uses a semi-automated region growing technique.

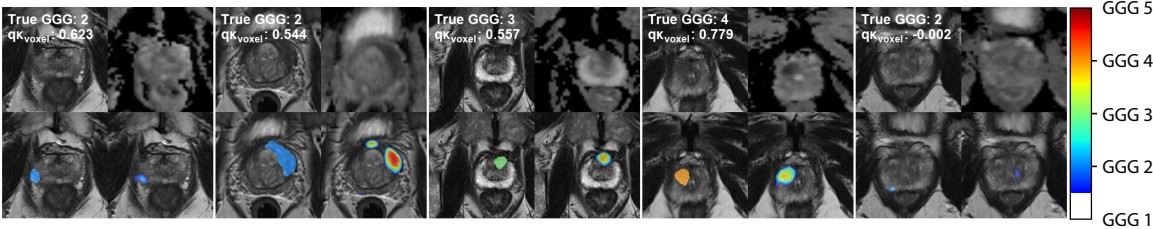

Figure 1: Qualitative results on the validation set using SLOR. Every image in the same 2×2 block represents the same middle slice of one lesion. The top right image is ADC, the other three are T2-w images. The bottom left overlay is the manual segmentation, and the bottom right overlay is the network prediction. Colormaps are transparent for $GGG$ 1, as this is clinically equivalent to healthy tissue. $q\kappa_{voxel}$ is calculated on the displayed slice.

**Pre-processing and CNN model:** We cropped the images around the image center, such that the entire prostate and lesions were included in this region of interest (ROI). All ROIs were $90 \times 90 \times 80$ $mm^3$ and resized to $192 \times 192 \times 32$ voxels. 2D slices of the T2-w and ADC ROIs were the two input channels of the model. The T2-w images were normalized per individual image. As ADC images are quantitative, they were normalized over the entire dataset. We used the following data augmentation: elastic deformation, contrast changing, rotation, shearing, flipping along the vertical axis. We used a U-Net (Ronneberger et al., 2015) with batch normalization (Ioffe and Szegedy, 2015) after each convolutional layer.

**Methods and experiments:** Cross entropy loss and Adam were used in all experiments. We evaluated three different approaches for incorporating $GGG$ scoring into the model. The last two models differ from the first one in that they take into account the ordinal nature of the problem. For example, if a lesion has $GGG$ 4, the first method penalizes predictions of 1 and 3 equally, while the last two approaches penalize the former more than the latter. The size of each output channel in all three methods was the same as the same size of the input images. *Multiclass classification (MCC):* The network had five output channels, where each output channel corresponded to a $GGG$. *Multi-label ordinal regression (MLOR):* The network, as described by (Niu et al., 2016; Cheng et al., 2008), had 4 output channels, where the $n^{\text{th}}$ channel was 1 if the $GGG$ is greater than $n$ and 0 otherwise. *Soft-label ordinal regression (SLOR):* The target in this method was a single channel wherein each voxel had a scalar value in the range of 0 and 1. For every tumor $T$, the voxel within that tumor $T$ had a value of $\frac{1}{4}(GGG_T - 1)$, where $GGG_T \in \{1, \ldots, 5\}$ is the $GGG$ of tumor $T$. This method takes into account the ordinal nature of the problem, since the loss increases as the difference between a prediction and target value increases.

**Evaluation:** In this work, we developed a method that also detects the location of lesions, instead of simply grading, which is the aim of PX-2. The validation was done using 5-fold cross validation. To evaluate the model on the challenge test set, we also calculated lesion-wise $GGG$ predictions for the coordinates given by the challenge. This was done by thresholding the prediction map ($GGG \geq 2$) and considering the connected components in a radius $r$ of the requested coordinate. If no connected component was present in a radius $r$ of the requested coordinate, a $GGG$ of 1 was assigned to that lesion. Otherwise,

Table 1: Results of different approaches. Metrics are expressed as $\mu \pm \sigma$ of the different cross-validation folds. $q\kappa_{voxel}$ is the voxel-wise quadratic-weighted $\kappa$ score. DSC is the Dice score after thresholding the prediction maps and target images with $GGG \geq 2$.

| Method | $q\kappa_{voxel}$ | DSC ($GGG \geq 2$) |
|--------|-------------------|--------------------|
| MCC | $0.225 \pm 0.114$ | $0.221 \pm 0.104$ |
| MLOR | $0.348 \pm 0.050$ | $0.303 \pm 0.059$ |
| **SLOR** | $\mathbf{0.391 \pm 0.062}$ | $\mathbf{0.321 \pm 0.039}$ |

$f_{prob \to GGG}(P_r)$ was used to predict the $GGG$, where $P_r$ is the list of all the voxel values in the connected component within radius $r$ of the requested coordinate and $f_{prob \to GGG}$ is a statistical measure. If the selection of the connected component was ambiguous, the largest connected component was chosen. We did a grid search by checking all permutations of $f_{prob \to GGG} \in \{90^{th}\ percentile, mean, max\}$ and $r \in \{0, 5, 10, 20, 30\}\ mm$. The input and predictions were not masked with a prostate segmentation.

## 3. Results

In Table 1, performance metrics of the different methods on the validation set are displayed. Fig. 1 shows randomly selected qualitative results of the SLOR approach. The grid search for $f_{prob \to GGG}$ and $r$ showed the highest performance when using the $90^{th}$ percentile for $f_{prob \to GGG}$ and choosing $5\ mm$ for $r$. The lesion-wise quadratic-weighted $\kappa$ ($q\kappa_{lesion}$) was $0.201 \pm 0.257$ ($\mu \pm \sigma$ of the 5-fold cross-validation iterations) using this method. On the PX-2 test set, which is not publicly available, SLOR had a performance of $q\kappa_{lesion} = 0.082 \pm 0.272$ ($\mu \pm \sigma$ calculated by the challenge organizers using bootstrapping with 1k iterations).

## 4. Discussion

We showed that the two ordinal regression approaches outperformed MCC. Furthermore, our proposed SLOR method was shown to perform better than MLOR, which is commonly used in literature for ordinal regression (Niu et al., 2016; Cheng et al., 2008). When converting the voxel-wise to lesion-wise predictions, $q\kappa_{lesion}$ drops. This could, for instance, be due to the method of this conversion, despite an extensive grid search. This could also be a result of small lesions being missed, which is not well reflected in the voxel-wise metric. The relatively low $\kappa$ scores can be explained by the inherent ambiguity of this problem. Pathologists, even when grading from tissue biopsies, score an unweighted $\kappa$ for inter- and intra-agreement of 0.54 and 0.66, respectively (Melia et al., 2006). Nevertheless, our grading method outperforms 30 out of 43 official submissions to PX-2, while solving a more difficult, but more clinically relevant, problem. The input was not masked with a prostate segmentation in our method, as that might exclude tumors that grow beyond the prostate border and essential information about the tumor aggressiveness might be missed. In future work we will explore the potential effect of including a prostate segmentation in other ways.

## Acknowledgments

We thank the ProstateX-2 challenge organizers for providing the data used in this research, and for calculating the test results of our submission.

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
