# OpenReview forum: "Simultaneous Detection and Grading of Prostate Cancer in Multi-Parametric MRI"
_MIDL.io/2019/Conference/Abstract — MIDL Abstract 2019_

### Official Review · AnonReviewer2 · 2019-04-28
**improve originality (beyond reusing existing architecture), methodological description (key details are missing on ordinal nature), and results (comparative study to appreciate the potential improvements)**

**Rating:** 2
**Confidence:** 3

**Review:**

Simultaneous Detection and Grading of Prostate Cancer in Multi-Parametric MRI

The paper uses CNNs to detect and grade prostate cancer lesion on mpMRI images. The method uses a U-net architecture to classify over multiple grades of lesions. The results indicate mitigated results with low kappa and dice scores.
The paper itself may benefit from further clarification on 1) originality (beyond reusing existing cnn architectures), 2) methodological description (key details are missing, notably on the ordinal nature of the problem, what is exactly learnt). Results should be compared with an existing literature to judge if there are improvements or not.

---

### Official Review · AnonReviewer1 · 2019-04-30
**Good paper with strong methodology**

**Rating:** 3
**Confidence:** 2

**Review:**

This paper is about both segmenting and grading prostate tumors (GGG, from 1 to 5). This is not trivial, as there is many methods that could do that (pixel-wise regression, one probability per grade, multi-class predictions, ..), and then different ways to aggregate the pixel-wise predictions. Therefore, a thorough evaluation is required.

The authors compare three methods to supervise and predict the tumor grade, pixel-wise:
- Multi-class classification
- Multi-class ordered classification: class n being true means that GGG > n
- One regression per class (from 0 to 1), scaled to have a continuous range [1, 5]

Methods to aggregate the pixel-wise predictions (to provide the final grading) are also compared (90th percentile, mean, max), as well as the radius of connected components (to limit noisy predictions).

Pros:
- The evaluation is thorough and the methodology is strong
- One strength of the method is the interpretability: one can easily see what the network detected as a tumor with its grade

---

### Decision · Program_Chairs · 2019-05-06
**Acceptance Decision**

Accept